# “Seeing and Being Seen” or Just “Seeing” in a Smart Classroom Context When Videoconferencing: A User Experience-Based Qualitative Research on the Use of Cameras

**DOI:** 10.3390/ijerph19159615

**Published:** 2022-08-04

**Authors:** Josep Petchamé, Ignasi Iriondo, Garazi Azanza

**Affiliations:** 1Department of Engineering, Universitat Ramon Llull (URL), La Salle, 08022 Barcelona, Spain; 2Department of Social Sciences, University of Deusto, Avenida de las Universidad, 24, 48007 Bilbao, Spain

**Keywords:** cognitive load, COVID-19 pandemic, Emergency Remote Teaching, face-to-face, smart classroom, videoconferencing, user experience, well-being

## Abstract

This research examines the form in which undergraduates use video cameras during videoconferencing class sessions in a Smart Classroom context and, more specifically, the reasons why a considerable number of students opt to turn off their cameras when videoconferencing during the sessions while others keep them on. The study was carried out in an institution that had previously switched from face-to-face teaching to an Emergency Remote Teaching solution, initially adopted in 2019–2020 to cope with the COVID-19 pandemic restrictions. Findings suggest that using cameras when videoconferencing is associated with increasing and enhancing the interaction between the student and the rest of the class, although not all students agreed with this conclusion. In addition, having the video cameras switched fomented socialization and improved the overall online learning experience for students. However, the surveyed students pointed out diverse negative elements, such as why they had to turn on their personal cameras, privacy concerns, and limitations derived from the available technology infrastructure, in addition to other factors such as distractions, anxiety, and cognitive load. This work discusses how these elements can influence the well-being and the user experience of the students, both positively and negatively.

## 1. Introduction

The outbreak of the **Co**rona**vi**rus **D**isease 2019 (COVID-19) pandemic that outburst in March 2020 [1] strongly impacted people’s lives, including teaching and learning activities [2,3] at all stages of education. Study programs and methodologies were modified [4,5] to face the restrictions imposed and to guarantee learning outcomes. Technology played a key role since it allowed the implementation of remote learning solutions [6,7,8], thus allowing the continuity of the academic activity, although this did negatively affect motivation, reduce the time spent on learning, lower levels of interaction, and increase the level of stress [9].

A considerable number of research works have focused on analyzing various aspects of the policies adopted to fight the pandemic and the effects that these measures have had on people in different facets of their lives [10,11,12]. This work focuses on how the use of cameras affected university students in the context of a Smart Classroom deployment during the COVID-19 pandemic. The Smart Classroom (SC) technology was introduced at La Salle Universitat Ramon Llull (La Salle-URL) in 2020–2021 [13,14] as an evolution and improvement of the Emergency Remote Teaching (ERT) strategy previously deployed in the Institution [15]. Both ERT and SC were implemented thanks to the ICTs (Information and Telecommunication Technologies) available. Initially, the ERT solution was implemented to cope with the prohibition of all physical access to classrooms and laboratories. Then, in three consecutive semesters, the students’ instruction switched from face-to-face (F2F) to ERT and later from ERT to SC. The SC solution that was later deployed included the potential use of a videoconferencing system [13], which implied the use of cameras. Class sessions were recorded, and students had a week to access its content through a streaming platform. In addition to the broadcasting possibilities that it offers, SC technology facilitates the creation of a smart learning environment [16,17], which may foment, among other advantages, personalized learning [18].

One of the theoretical advantages of the SC system over the ERT format is that the installation of cameras and microphones deployed in the classrooms and the laboratories enabled live broadcasts of the class sessions. Furthermore, the SC solution that was implemented allowed participants who were attending classes off-campus to be seen and heard by the rest of the class thanks to the audio and video devices of their personal devices, typically a personal computer, used to connect to the class session. From our previous research works, students rated the different features offered by the SC format highly [15,19]. However, a high number of instructors at LaSalle-URL have complained that many students preferred to turn off their personal cameras while attending classes off-campus.

### 1.1. Previous Research Works about the Use of Cameras in Videoconferences

Different research works have focused on studying the feelings and the thoughts of the students on the use of cameras when videoconferencing. Videoconferencing is a technological solution implemented by a wide range of institutions that can be carried out by means of diverse online meeting platforms [20], such as Zoom, Cisco Webex Teams, Skype, Microsoft Teams, or Google Meet/Google Hangouts, platforms that are analyzed in diverse research studies, such as [21,22]. As posited in [23], videoconferencing can be taxonomized into three different categories according to its functionality: (1) Desktop Videoconferencing; (2) Interactive Videoconferencing; (3) Web-Based Videoconferencing. Despite their differences, out of the scope of this research, all the mentioned options are based on the use of cameras and microphones, which allow distinct levels of interaction through the images and the sounds of the participants. In this context, cameras and, more specifically, their use, has become a key element when thinking about the success of the user experience. Videoconferencing has proved to be a useful tool, among others, when researching to collect data [24], to carry out educational experiences [7,15,21,23,25,26], or simply as a mechanism to improve health at different levels [27,28] and promote well-being [29,30]. However, digital technologies are also the cause of negative consequences such as technology-related stress, overload, anxiety, interruption and distractions, addiction, etc. [31]. In fact, according to diverse research works, technostress, defined as the “stress experienced by the individual due to the use of ICTs” [32], when it occurs, has a negative impact on students’ well-being [33,34].

Recently, diverse issues have been raised in the research arena about the use of cameras when videoconferencing, such as: (1) why students turn their cameras on or off [25,26,35,36,37]; (2) the fatigue associated with videoconferencing [38,39,40,41,42,43,44] and how to cope with this issue [41,45,46,47] or the development of a conceptual model about this topic [48]; (3) engagement or disengagement [49,50,51], a key element when analyzing learning [52,53]; (4) difficulties in maintaining attention [41,54]; (5) the emotions that result from using cameras in synchronous learning [55], including the stress [38] and anxiety caused by videoconferencing [56,57,58]; (6) privacy concerns [36,59,60,61]; (7) users’ preferences and comparisons when using F2F versus online formats [15,62,63]; or (8) guidelines and recommendations for users when videoconferencing [20,21,49,64], and even in a broader scope, analyzing on line learning through literature review approaches [65,66], or examining remote teaching in terms of its strengths, weaknesses, opportunities, and challenges [67].

Having analyzed the use of cameras when videoconferencing, different research works have found that users experience an increase in cognitive load as a consequence of the use of cameras [43,68,69,70]. Cognitive-load theory (CLT) studies the cognition mechanisms that human beings have, and one of its goals seeks to optimize learning [71,72]. CLT is based on the fact that people have a limited working memory which results in a processing limit for information at any one time [72,73], which implies dealing with a maximum cognitive load. According to [72], different categories conform to the cognitive load: (1) intrinsic, i.e., the complexity or difficulty of the information to process; (2) extraneous, i.e., how the information is presented or other factors linked to the instruction procedure that is actually used; (3) germane, i.e., which resources are needed to learn. In fact, factors related to the environment, such as the way the teaching materials are constructed, the physical space where students are taught, and even the other participants involved in the instruction process, have an impact on the cognitive load that students have to deal with [74].

Students’ well-being can be defined as “a population-based term targeting positive feelings about oneself and reflecting an inner capacity—a resourcefulness—to deal with the pressures and challenges of student life and learning” [75]. In fact, individual differences drive differences in the perception of well-being [76]. Furthermore, when referring specifically to the impact that technology has on well-being, we talk about digital well-being [77]. Well-being deals with different elements, such as physical health and mental health, the latter including various positive and negative emotions, such as happiness, satisfaction, anxiety, stress, depression, and so on [78], affecting both instructors [79] and students. Well-being is a core topic when considering the impact of technology [77] since, despite its leverage on teaching and learning, it also has an impact, among others, on personal intrusion and stress [80] or on psychological benefits and costs [81]. Different online learning experiences driven during the COVID-19, such as ZOOM cohort chats [82] adopted during the COVID-19 crisis, have had a positive effect on students’ well-being and student satisfaction levels [83]. However, face-to-face contact seems to benefit well-being more than computer-mediated interactions, as posited in [63].

### 1.2. Purpose and Goals of This Research Work

The aim of this research work is to shed light on the undergraduates’ feelings about the use of cameras in the videoconferencing system through a qualitative approach. To perform this exploratory research about the use of cameras, a User Experience (UX) approach was applied to shed light on the topic. Since ICT engineering undergraduates are sufficiently digitally capable, this work does not include areas such as reluctance, fear, and the potential incompetence associated with the use of ICT technology.

## 2. Materials and Methods

This research has been carried out by surveying second-year ICT engineering students who experienced a Smart Classroom system during their first year at the university [13,14], which enabled them to study remotely during the COVID-19 pandemic.

### 2.1. Participants and Procedure

The participants were engineering undergraduates from La Salle-URL enrolled in seven different ICTs specialties who attended the second-year annual subject ‘Value Chain and Financial Economics’ in the 2021–2022 academic course. The total number of enrolled students in the subject was 133, and 85 of the students that attended the last session of the class of the fall term were invited to complete an open-ended questionnaire voluntarily and anonymously on the use of cameras through a pocket **B**ipolar **La**ddering (BLA) assessment tool, as detailed in Section 2.2. The question that the undergraduates were asked was: “Based on your experience of remote classes through videoconferencing, which are your perceptions of using cameras?”. The survey was administered through a google-forms questionnaire and was properly answered by 79 students (92.94% of the attending students). Ages ranged from 18 to 25 years old (M = 19.58; SD = 1.56), and in terms of gender the sample included 16 females (20.75%) and 62 males (79.75%).

The sample of respondents can be considered homogeneous since (1) all of the students had physically attended classes on campus, (2) all of the students attended class sessions by a videoconferencing system through an SC deployment on campus, and (3) all of the surveyed undergraduates were studying an ICT engineering degree.

The qualitative research was planned following the guiding principles of other research works [84,85,86,87,88,89]. Diverse steps were followed once the aim of the research had been determined, as shown in [84]: (1) To identify the potential respondents; (2) To decide the methods to collect data; (3) To select the analysis methodology to be performed. Data were collected from undergraduates through a pocket BLA tool, a homogeneous sample compounded of two-year ICT students. The collected items were analyzed individually by three researchers, who finally reached a consensus once they compared their individual findings. The latter process constitutes a triangulation, which enhances the validity of the findings [84,90].

### 2.2. Cameras on or Cameras off: Students’ UX When Videoconferencing through a BLA Tool

UX enables the collection of feedback from the users once they have experienced a product or a service [91], being the user at the center of the assessment process [92]. Diverse research studies have used the UX approach to assess students’ perceptions in educational contexts, such as [15,19,93,94,95]. In this research work to conduct the UX assessment through a qualitative approach, the pocket BLA tool was used to collect the students’ feedback on the use of cameras when videoconferencing, as carried out in other research studies in the educational field [15,19,96]. There are two modalities of BLA: (1) administering the instrument by a face-to-face interview, which is known as a BLA tool; (2) conducting a written survey, named the pocket BLA tool. On the one hand, the first alternative allows for the clarification of the collected answers, despite requiring a lot of time since the interviewer must hear a single person during the interview. On the other hand, the pocket BLA tool allows for the collection of a substantial number of answers in a relatively brief period since all of the questionnaires can be answered at the same time by the surveyed people.

BLA usage has been described in various research works, e.g., [15,19], and is based on a Socratic approach, since respondents are required to give their opinion from just a single open-ended question. So, if the question is properly formulated, there are no biases or influences in the collected answers since the interviewed people fulfill a *tabula rasa*, in Socratic words [97]. The mechanics of the BLA tool is explained as follows: the participants are asked to give their feelings about the different positive and negative elements that they identify once they have concluded an experience as users. Through the BLA tool, different elements of the user experience are named through a description and scored from 0 (lowest possible level of satisfaction) to 10 (maximum level of satisfaction), depending on the experience that each user has lived. According to [97], values close to 0 show that users find the referenced element very uncomfortable or unpleasant, and if it is highly cited, it can be considered a warning that the item needs improvement. However, values close to 10 indicate that users consider an element to be very comfortable or pleasant, and therefore, this item could remain the same. In addition, it can be useful to know the level of heterogeneity of an element since if a highly cited element has very different marks, we can think that it affects the users in a very different way. In addition, the user is asked to propose ideas that could increase their satisfaction with each one of the mentioned positive and negative elements. When talking about analyzing the answers, both the description of the element and its justification, plus the proposed improvement, constitute a single pack that defines each element clearly. Hence, once all of the participants have given their feedback, all the elements are presented, classified, and studied depending on how many times every single element is cited, which is presented by means of a ‘Mention Index’ of each element. Then, there are elements that are mentioned by several users, named ‘Common Elements’, while others are cited by just one participant; each one of them is referred to as a ‘Particular Element’. The template of the pocket BLA that has been presented to the surveyed student is shown in Figure 1.

## 3. Findings

Table 1 shows all of the positive elements found by the undergraduates once their opinions about the use of cameras in a smart classroom videoconferencing context during the 2021–2022 academic year had been analyzed and categorized. The items with the label starting with the letters PcE name a positive element according to two or more students, while those with PpE name an element that has only been mentioned by a single undergraduate. The mention index (labeled as ‘Mention I.’ in Table 1) shows the number of students that have cited a specific element, while the average score provides an idea of the degree of user satisfaction with the item and the variance reflects the degree of heterogeneity.

Table 2 shows all of the negative elements cited by the students according to their experience. Again, the items found by the undergraduates are grouped and displayed, ranged by their mention index.

As a result of the BLA tool, different proposals to improve each of the elements pointed out by the students were collected. Once all of the students’ contributions were analyzed by the researchers, the most notable are listed as follows. The most repeated improvement was to remove the compulsory use of cameras when attending class sessions through videoconferencing when referring to different elements: PcE_04, PcE_11, NcE_01, NcE_07, NcE_13, and NcE_14. In this line, different ideas mentioned by the students are listed next: “Do not make the use of cameras when attending classes off campus compulsory”; “Not mandatory to switch cameras on”; “Allow the student to choose whether to switch on their camera or not”; “Do not force students to switch on the cameras; displaying the name of the student would be enough instead of showing his image”; “Convince the student to switch on the camera, instead of making it compulsory”, “Flexibility in camera use” or “Non-mandatory use of the students’ camera except in evaluations”. However, other students emphasized the idea that if connecting the camera was compulsory, instructors should be strict in demanding that all the students had their cameras on, as shown in the following opinions: “Force the student who is speaking to switch their camera on”; “Make all students have their cameras switched on” or “Force the student to have the camera activated at all times, except on specific occasions”.

The students also detailed diverse opinions about how to improve interaction issues when referring to PcE_01, PcE_02_PcE_11, and NcE_15. Here, different thoughts may be mentioned as follows: “Improve interaction”; “More interaction”; “Improve interaction techniques by creating small and more participatory groups”; “Make class sessions more participatory”; “Include more interaction with students” or “Increase interaction with off-campus students”. When referring to privacy concerns, the students pointed out: “Despite allowing class sessions to be followed well, different mechanisms should be enabled to preserve privacy” or “Having the camera switched on can cause privacy issues when other colleagues see your home or take screenshots; it should be avoided”. To cope with distractions, “Make class sessions more participatory”, “Make class sessions more pleasant and fun, so that the student has an enjoyable time and thus avoids distractions”, or “Make the sessions of class much more enjoyable”. Students assessed the SC formats PcE_04, PcE_11, and PpE_06 to be very satisfactory, as shown in the following: “Take more advantage of the benefits offered by cameras” or “The off-campus format has been very useful during the confinement, it is an option that should be maintained in the future”. However, other students preferred the face-to-face format: “Make the class sessions in face-to-face format”. Being able to see the recorded class sessions was highly appreciated, and suggestions to improve that choice were: “Make recorded classes accessible for a month, instead of just for a week”; “Continue with the practice of recording class sessions in the future” or “Keep the recordings of the class sessions since they enable students to review concepts that have not been understood”.

## 4. Discussion

This research was carried out on students who had both attended classes on campus and been taught using the online format during their first year at the university. On the one hand, classrooms and laboratories were adapted to facilitate F2F teaching on campus [15]. On the other hand, if a student or anyone he or she had been in close contact with had experienced symptoms or tested positive for COVID-19, he or she was prohibited from physically attending classes on campus and had to quarantine for at least 10 days. In the latter case, students could keep on attending online class sessions via the Smart Classroom system through a Zoom-based videoconferencing system [13,14,15]. A previous study was carried out comparing F2F, ERT, and SC [15], while this qualitative research work focuses specifically on the students’ perceptions of the use of cameras when attending class sessions.

Strong student-to-student and student-to-instructor interaction top the ranking of the list of positive elements highlighted by the undergraduates surveyed in terms of mention index while appearing as a negative element for a group of students in line with [9,36]. This dichotomy in terms of perception could be explained depending on what were the students’ references when comparing SC: ERT or F2F, as shown in [15]. Higher levels of concentration and being more attentive during the class sessions are two items highly appreciated by a considerable number of students. However, a substantial number of respondents complain about the exposure to a lot of distractions, in line with [31,67]. The fourth most appreciated element by students was not explicitly connected with the use of cameras but with the format as a choice in front of the F2F sessions: Flexibility, in line with [67]; and comfort, thanks to the possibility of attending classes online. In the same line, a range of students emphasized that attending classes through the use of cameras offered an experience that was very close to attending classes physically in the campus facilities, in line with other research works [15]; though other students stated explicitly that the videoconferencing system was not as good as the F2F format. Students were satisfied with the access to recorded class sessions that allowed them to watch a previous session again to clarify concepts or just to watch missed classes, although several students showed their reluctance to be recorded. It should be noted that the possibility of watching recorded class sessions (PcE_08) was the element that was assessed with the highest value, a 9.2 in terms of mean when comparing the scores of the different positive common elements identified by the students once they had experienced the use of cameras. Other items pointed out by the students involved making classes more dynamic, more motivating, and more personalized, all of them assessed with high mean values in terms of satisfaction.

The most mentioned negative item was the obligation to switch cameras on when attending classes off campus, cited by twenty students. A research study found that students used their cameras at the start of the class sessions in order to solve their socio-affective needs, then turned off their cameras after some sessions [98]. In different research studies, the use of cameras was not compulsory for the students that were attending their synchronous classes through a videoconferencing format, e.g., [25,37]. Difficulties in maintaining attention when they were off campus were cited by the students, which was consistent with the findings of other studies [41]. Students complained about diverse technical issues in line with other research works, e.g., [25,67], such as the limitations derived from being connected to the internet, the poor image resolution of the cameras, poor audio quality, or even the fact that some personal computers do not have cameras. However, none of the concerns were related to technical issues, such as the use of cameras or even the installation of external web cameras when needed, which confirms the first assumption that, given that the surveyed students were ICTs engineering undergraduates, they were digitally adept and had sufficient experience of working with cameras. Therefore, they did not consider technology itself a problem. Privacy concerns derived from the use of cameras were mentioned by a significant number of those surveyed, in line with other research works, such as [25,35,36,37], and were consistent with the concerns derived from the use of videoconferencing [59,60,61]. More specifically, when talking about their privacy concerns, the students pointed out diverse comments as follows: “Many students do not want to show their personal space”; “The camera show your house to unknown people”; “it is invasive”; “violates your privacy”; “you cannot totally control what happens in your place of study” or “you lose privacy since the whole world can see your room and what you do”.

Our findings suggest the presence of cognitive load. Among the most common negative elements, we found items related to computer slowdown (NcE_02), distractions (NcE_04), poor audio quality (NcE_08), less instructors’ availability (NcE_10), and difficulty in reading what is written on the blackboard (NcE_11). These elements may hinder understanding and increase the students’ feeling of complexity, unclarity, and ineffectiveness of instructions and explanations, which have been previously identified as components of cognitive load [99]. However, positive items such as “More enjoyable class sessions: Quality, understanding” (PcE_11) suggest that the use of cameras may also increase understanding and thus reduce cognitive load. Future research is needed to more specifically address the relationship by including include the use of specific instruments to measure cognitive load, such as the Multidimensional Cognitive Load Scale for Virtual Environments developed by Andersen and Makransky [99].

Regarding student well-being, among the most mentioned items, we found those referring to interaction and socialization (PcE_01 and PcE_03), flexibility and comfort (PcE_04), a feeling of better communication (PcE_05), and more enjoyable class sessions (PcE_11), among others, which may positively affect students’ satisfaction and may be related to lower levels of anxiety. However, other items such as “Feeling of being watched” (NcE_07) and “Shame, insecurity, feeling uncomfortable” (NcE_13) suggest that the use of cameras may also increase dissatisfaction and anxiety or stress levels. Differences in the perception of well-being have been previously explained by individual differences [76], and therefore, future research should explore the role of students’ personalities in the impact of the use of cameras on well-being.

The limitations of this research work are presented in the following. Firstly, the undergraduates surveyed are engineering students who are tech-savvy in ICTs topics and, therefore, less likely to suffer from the issues associated with the use of the analyzed technology. This may mask concerns about the use of cameras by less digitally-adept users. To remedy this, further studies using other profiles that are not specifically oriented to technology could be carried out to generalize the findings on the use of cameras. Secondly, since all of the responses to the single open-ended question were requested to be given in a short format, some of the comments and suggestions given by the students were limited and most of the answers used less than fifteen words. Once the findings are identified, a qualitative survey can be performed to quantify the elements that have been mentioned by the student. Finally, as this study was conducted using second-year students, expanding this survey to first-, third-, and fourth-year students would enable us to check if the perceptions about the use of cameras in the analyzed context vary depending on the year of study. As far as further research directions are concerned, this UX-Based qualitative work on the use of cameras when videoconferencing in a tech-oriented undergraduates’ context may lead to diverse research lines. On the one hand, by replicating this study to non-tech-oriented students, on the other hand, by conducting quantitative research work to shed light on this research topic.

## 5. Conclusions

Various conclusions can be drawn from the analysis of the data collected from the survey to capture student perceptions on the use of cameras when videoconferencing in a Smart Classroom system. The fact that respondents highly rate the option of being able to see their instructors and classmates in real-time through the cameras when videoconferencing is reflected in two main outcomes: better interaction, despite the fact that this element was considered a negative element for a smaller group of students; and more socialization. Moreover, the students stressed that thanks to the videoconferencing system based on the use of cameras, they have a high degree of flexibility in terms of class attendance as they can choose whether to take classes on campus or remotely. However, many students said that in terms of interaction, they preferred the F2F to the SC format and requested the implementation of mechanisms to improve interaction with other online students, which was consistent with previous findings presented in [15]. With respect to the negative items, a significant number of students were reluctant to connect their cameras when they were off campus. The other most common negative items are as follows: problems derived from the internet connection, privacy concerns, and the distractions that students face when attending classes online. To sum up, although the use of cameras when videoconferencing is an element that improves online classes according to student perceptions, its design in terms of interaction, content, and dynamics should be carefully planned to avoid a negative impact on student well-being.

## Figures and Tables

**Figure 1 ijerph-19-09615-f001:**
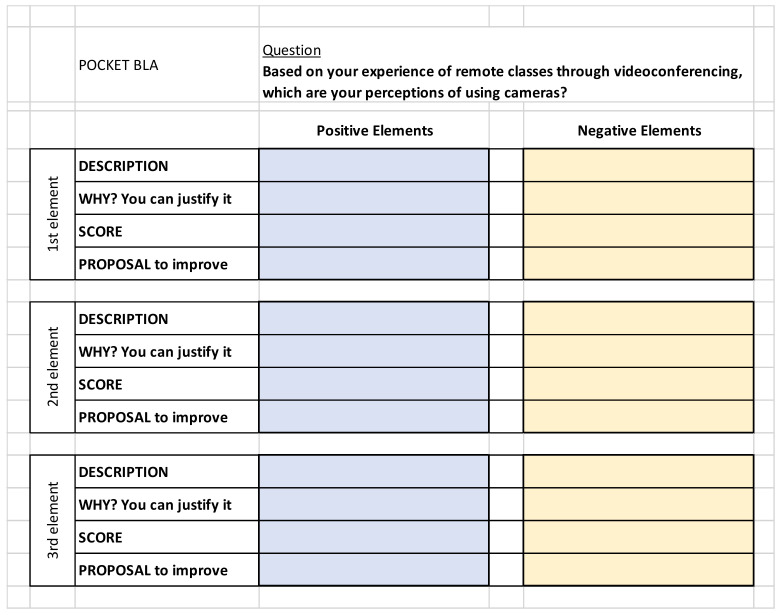
Pocket BLA: Template to be completed by the participants about the use of cameras when videoconferencing.

**Table 1 ijerph-19-09615-t001:** Positive (P) common (c) and particular (p) Elements.

Item	Description	Average	Variance	Mention I.
PcE_01	Better interaction with instructors and classmates	7.9	1.5	27/79
PcE_02	You feel more compelled to be attentive and to concentrate	7.5	1.1	26/79
PcE_03	Better socialization among students/Seeing other classmates	7.2	0.9	26/79
PcE_04	Allows for off-campus class sessions: Flexibility & comfort	8.0	1.1	23/79
PcE_05	Being able to follow the instructor via camera from home	8.1	1.8	17/79
PcE_06	Feeling of better communication	8.4	1.0	16/79
PcE_07	The instructors get better feedback from the students	8.0	1.3	15/79
PcE_08	Being able to see recorded class sessions	9.2	1.4	12/79
PcE_09	It forces us to be present at class	7.4	0.2	11/79
PcE_10	More sense of presence in virtual class sessions/Like F2F	8.3	1.1	9/79
PcE_11	More enjoyable class sessions: Quality, understanding	8.3	1.6	7/79
PcE_12	You can put a background behind you to respect your privacy	8.4	1.0	5/79
PcE_13	It makes the class session more personal	7.6	0.2	5/79
PcE_14	Instructors are more motivated and eager to teach	8.5	1.5	2/79
PcE_15	Better whiteboard visibility than when being in the classroom	8.5	0.3	2/79
PcE_16	The camera promotes more dynamic class sessions	8.0	0.0	2/79
PcE_17	Cameras and microphones allow a good immersion in the class	7.5	0.3	2/79
PpE_01	You can share the screen to solve doubts	9.0	-	1/79
PpE_02	Helps to generate routine: obligation to dress appropriately, …	9.0	-	1/79
PpE_03	Facilitates the monitoring of class sessions via Smart Classrooms	8.0	-	1/79
PpE_04	Allow virtual background to prevent messy room from being seen	8.0	-	1/79
PpE_05	You do not feel the pressure of being in a classroom	7.0	-	1/79
PpE_06	More familiar than a black and white picture/ERT	7.0	-	1/79
PpE_07	You can see and hear everything that happens in the classroom	7.0	-	1/79
PpE_08	More willing to use the cameras if other classmates use cameras	7.0	-	1/79
PpE_09	You can see what others are doing without interrupting	7.0	-	1/79
PpE_10	Enables attending class sessions without wearing a facemask	7.0	-	1/79

User Experience: Findings from the pocket BLA tool. Positive elements. Sorted by Mention Index.

**Table 2 ijerph-19-09615-t002:** Negative (P) common (c) and particular (p) Elements.

Item	Description	Average	Variance	Mention I.
NcE_01	Obligation to connect the camera when being off campus	3.4	2.1	20/79
NcE_02	Internet connection: Limitations and computer slowdown	3.3	1.6	14/79
NcE_03	Privacy concerns	3.2	2.3	14/79
NcE_04	There are many distractions	4.0	1.4	10/79
NcE_05	There are computers that do not have cameras	3.6	0.8	9/79
NcE_06	Poor image resolution of the cameras	3.5	2.5	8/79
NcE_07	Feeling of being watched	3.1	1.8	8/79
NcE_08	Audio issues: poor audio quality	4.1	0.4	7/79
NcE_09	Difficulty of staying focused	3.6	0.2	6/79
NcE_10	Less instructors’ availability, compared to on campus F2F option	3.1	1.1	6/79
NcE_11	Sometimes it is difficult to see what is written on the blackboard	4.0	2.4	5/79
NcE_12	Fixed appearance on the class screen of the last student who talked	3.8	0.5	5/79
NcE_13	Shame, insecurity, feeling uncomfortable	3.4	1.0	5/79
NcE_14	It is difficult to take part when being in front of the camera	3.2	1.7	5/79
NcE_15	There is not enough interaction with the other off campus students	3.5	0.3	4/79
NcE_16	You cannot be stretched out in bed/It forces you to be presentable	2.7	2.2	4/79
NcE_17	Greater difficulty to communicate with the instructors	3.6	0.2	3/79
NcE_18	Being recorded	3.6	1.5	3/79
NcE_19	You do not always want to see your face as close as it shows	2.3	1.6	3/79
NcE_20	You need for a quiet environment	5.0	0.0	2/79
NcE_21	It is different from F2F class attendance	4.5	0.2	2/79
NpE_01	Not helpful	4.0	-	1/79
NpE_02	Sudden zooming of cameras located in the classrooms	4.0	-	1/79
NpE_03	The instructor may assume that he or she is not heard	4.0	-	1/79
NpE_04	The instructor does not realize of what is written in the chat	4.0	-	1/79
NpE_05	Excessive control of attendance at the class sessions	3.0	-	1/79
NpE_06	Not all the students connect their cameras	3.0	-	1/79
NpE_07	Requirement of authorization to connect to the session of class	3.0	-	1/79
NpE_08	You do not see well the face of the other students	3.0	-	1/79
NpE_09	The teaching staff are more focused on F2F students	1.0	-	1/79
NpE_10	Difficulty to distinguish who is on campus or off campus	1.0	-	1/79

User Experience: Findings from the pocket BLA tool. Negative elements. Sorted by Mention Index.

## Data Availability

Data available on request from the authors.

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
