# Peer review of "“Seeing and Being Seen” or Just “Seeing” in a Smart Classroom Context When Videoconferencing: A User Experience-Based Qualitative Research on the Use of Cameras"

_ijerph, 2022, doi:10.3390/ijerph19159615_

Round 1

Reviewer 1 Report

The authors deal with research in the field of education. They focused on the positives and negatives of the turned on camera during online teaching. I consider the topic to be original and it is necessary to address it, as a university teacher I have had a lot of my own experience with online teaching in the last period.

Setting up a methodology for research yields results, but I would consider processing the results more accurately. Perhaps the solution would be to categorize the responses with statistical processing with respect to the impact factor of the journal.  There is a duplication of data in lines 264,265

On the other hand, I have a positive perception of the qualitative aspects of the work, as real statements of students can have a more significant impact on the learning process than statistical results.  

I rate the sources used as appropriate and the citations in terms of content correspond to the use in the text. 

Author Response

Thank you for your comments and suggestions. We give our answers attached to your comments.

The authors deal with research in the field of education. They focused on the positives and negatives of the turned on camera during online teaching. I consider the topic to be original and it is necessary to address it, as a university teacher I have had a lot of my own experience with online teaching in the last period.
Setting up a methodology for research yields results, but I would consider processing the results more accurately. Perhaps the solution would be to categorize the responses with statistical processing with respect to the impact factor of the journal. 

We have included some additional content in subsection 2.2. We have explained with more detail the BLA and the pocket BLA tool. We have also included a new reference that was indirectly cited in the previous version of the paper.

  1. Pifarré, M.; Tomico, O. Bipolar laddering (BLA): a participatory subjective exploration method on user experience. In Proceedings of the Proceedings of the 2007 Conference on Designing for User eXperiences; 2007; pp. 2–13.

The quantification of the different elements is just to stablish the importance of each element for each user, or what is the same, the quantification is just to rank the different elements. Besides, we have included a template of the survey presented to the students (Figure 1).

We have reworded different sentences and we have referenced explicitly the elements that the students have identified that are connected to the ‘cognitive load’ and to the “students’ well-being”.

We have included two new references to reinforce our explanation:

  1. Anglim, J.; Horwood, S.; Smillie, L.D.; Marrero, R.J.; Wood, J.K. Predicting psychological and subjective well-being from personality: A meta-analysis. Psychological Bulletin 2020, 146, doi:10.1037/bul0000226.
  2. Andersen, M.S.; Makransky, G. The validation and further development of a multidimensional cognitive load scale for virtual environments. Journal of Computer Assisted Learning 2021, 37, 183–196, doi:10.1111/jcal.12478.

There is a duplication of data in lines 264,265

The duplication issue was solved.

On the other hand, I have a positive perception of the qualitative aspects of the work, as real statements of students can have a more significant impact on the learning process than statistical results.  

I rate the sources used as appropriate and the citations in terms of content correspond to the use in the text. 

Reviewer 2 Report

The introduction is quite well written. There is a good connection with the literature review. The goal is presented in a clear manner, and it is mentioned that we will read a paper done with an exploratory design.

The methodology must clarify the BLA tool. We should read about it at 2.2, but the description is vague and needs improvement. Also, the BLA idea is to "quantify" different aspects of user experiences, so you should have a quantitative design and not a qualitative one. Please explain your option for the qualitative perspective!

Findings are quite plain and, in my opinion, need some improvements. We see only a "positive" and a "negative" list of elements. I consider that an article at this level should present some more findings. Even though we have an explorative study we should see more connections done with the data.

The discussions are connected with previous literature even though there are only very descriptive results.

Author Response

Thank you for your comments and suggestions. We give our answers attached to your comments.

The introduction is quite well written. There is a good connection with the literature review. The goal is presented in a clear manner, and it is mentioned that we will read a paper done with an exploratory design.

The methodology must clarify the BLA tool. We should read about it at 2.2, but the description is vague and needs improvement. Also, the BLA idea is to "quantify" different aspects of user experiences, so you should have a quantitative design and not a qualitative one. Please explain your option for the qualitative perspective!

We have included some additional content in subsection 2.2 to cope with your comment. We have explained with more detail the BLA and the pocket BLA tool according to your suggestion. We have also included a new reference that was indirectly cited in the previous version of the paper. The quantification of the different elements is just to stablish the importance of each element for each user, or what is the same, the quantification is just to rank the different elements. Besides, we have included a template of the survey presented to the students (Figure 1).

We have included a new reference to reinforce our explanation:

  1. Pifarré, M.; Tomico, O. Bipolar laddering (BLA): a participatory subjective exploration method on user experience. In Proceedings of the Proceedings of the 2007 Conference on Designing for User eXperiences; 2007; pp. 2–13.

Findings are quite plain and, in my opinion, need some improvements. We see only a "positive" and a "negative" list of elements. I consider that an article at this level should present some more findings. Even though we have an explorative study we should see more connections done with the data.

We have reworded different sentences and we have referenced explicitly the elements that the students have identified that are connected to the students’ perceptions, the ‘cognitive load’ issue and the “students’ well-being”.

We have included two new references to reinforce our explanation:

  1. Anglim, J.; Horwood, S.; Smillie, L.D.; Marrero, R.J.; Wood, J.K. Predicting psychological and subjective well-being from personality: A meta-analysis. Psychological Bulletin 2020, 146, doi:10.1037/bul0000226.
  2. Andersen, M.S.; Makransky, G. The validation and further development of a multidimensional cognitive load scale for virtual environments. Journal of Computer Assisted Learning 2021, 37, 183–196, doi:10.1111/jcal.12478.

The discussions are connected with previous literature even though there are only very descriptive results.

Reviewer 3 Report

Review of ijerph-1763086

“Seeing and Being Seen” or Just “Seeing” in a Smart Classroom 2 Context when Videoconferencing: A User Experience-Based 3 Qualitative Research on the Use of Cameras

 This manuscript starts with an informative introduction that gives a general overview of the context of the research. The research’s main question was “This work focuses on how the use of cameras affected university students in the context of a Smart Classroom deployment during the COVID-19 pandemic” (l. 39, 40). While the literature review is extensive and deals with interesting topics, they are not entirely developed in the rest of the manuscript. For instance, the authors presented the Cognitive Load Theory (CLT) and explained what it was, but it was lightly connected to the research they presented (“Many of the elements previously mentioned may have an impact on the students’ cognitive load” l. 238); it is not more than speculation as no hard data was presented. They also talk about technology’s impact on students' (and faculty's) well-being or the negative impacts it may produce. Alas, the research did not follow that line of inquiry, but it presented general statements (“may contribute to the students’ well-being” – l. 291). In a sense, the extensive and enlightening literature review seems disconnected from the research. Given that the main question is quite open-ended, the reader may expect that the study will cover the topics presented in the literature review.

In the course of the manuscript, there are several points that need more explanation. For one, the User Experience (UX) approach needs to be explained in a more comprehensive way, so the reader understands how this approach works and how it impacts the research study. A more thorough description of the pocket Bipolar Laddering (BLA) assessment tool will shed light on how the research was conducted. Is the BLA a piece of technology? Is it a questionnaire? The survey that was administered also needs to include more details. The authors stated that the survey asked students to respond to the question “Based on your experience of remote classes through videoconferencing, which are your perceptions of using cameras?” (l. 144), which is extremely vague. Later, the report gives the impression that there was more than one question in the survey (“Secondly, since all responses were requested to be given in a short format, some of the comments and suggestions given by the students were limited” – l. 300-301). Perhaps including a copy of the survey may contribute to the clarity of the manuscript.

In terms of analysis of data, the authors may want to use emergent themes as an inductive analysis that will allow them to present the results in a condensed format by merging similar responses. As the results are presented in the manuscript, there is too much repetition of responses to the point that it appears that the data was not analyzed using a qualitative approach. In fact, one of the most mentioned topics was that students disliked having their cameras on during video conferencing, but there is no explanation as to why the students felt that way other than “there were privacy concerns.” What did the students mean by that? There were some explanations presented in the literature review but not from the collected data. On the other hand, the conclusion, limitations of the study, and further research are clear and inspiring to conduct more research.

Author Response

Thank you for your comments and suggestions. We give our answers attached to your comments.

This manuscript starts with an informative introduction that gives a general overview of the context of the research. The research’s main question was “This work focuses on how the use of cameras affected university students in the context of a Smart Classroom deployment during the COVID-19 pandemic” (l. 39, 40). While the literature review is extensive and deals with interesting topics, they are not entirely developed in the rest of the manuscript. For instance, the authors presented the Cognitive Load Theory (CLT) and explained what it was, but it was lightly connected to the research they presented (“Many of the elements previously mentioned may have an impact on the students’ cognitive load” l. 238); it is not more than speculation as no hard data was presented.

They also talk about technology’s impact on students' (and faculty's) well-being or the negative impacts it may produce. Alas, the research did not follow that line of inquiry, but it presented general statements (“may contribute to the students’ well-being” – l. 291).

In a sense, the extensive and enlightening literature review seems disconnected from the research. Given that the main question is quite open-ended, the reader may expect that the study will cover the topics presented in the literature review.

We have reworded different sentences and we have referenced explicitly the elements that the students have identified that are connected to the ‘cognitive load’ and to the “students’ well-being”.

We have included two new references to reinforce our explanation:

  1. Anglim, J.; Horwood, S.; Smillie, L.D.; Marrero, R.J.; Wood, J.K. Predicting psychological and subjective well-being from personality: A meta-analysis. Psychological Bulletin 2020, 146, doi:10.1037/bul0000226.
  2. Andersen, M.S.; Makransky, G. The validation and further development of a multidimensional cognitive load scale for virtual environments. Journal of Computer Assisted Learning 2021, 37, 183–196, doi:10.1111/jcal.12478.

In the course of the manuscript, there are several points that need more explanation. For one, the User Experience (UX) approach needs to be explained in a more comprehensive way, so the reader understands how this approach works and how it impacts the research study. A more thorough description of the pocket Bipolar Laddering (BLA) assessment tool will shed light on how the research was conducted. Is the BLA a piece of technology? Is it a questionnaire? The survey that was administered also needs to include more details.

We have included some additional content in subsection 2.2 to cope with your comment. We have explained with more detail the BLA and the pocket BLA tool according to your suggestion. We have also included a new reference that was indirectly cited in the previous version of the paper.

The authors stated that the survey asked students to respond to the question “Based on your experience of remote classes through videoconferencing, which are your perceptions of using cameras?” (l. 144), which is extremely vague.

The BLA tool is designed to collect the user experience about a product or a service with the aim of improving it from the user’s opinions. It is important to offer a ‘tabula rasa’, or what is the same, proposing an open question, the clearer as possible, but avoiding, if possible, any bias. Once you have identified the positive and negative elements from scratch, a second step is to focus some specific elements (by means of qualitative or quantitative surveys).

We have included a new reference to reinforce our explanation:

  1. Pifarré, M.; Tomico, O. Bipolar laddering (BLA): a participatory subjective exploration method on user experience. In Proceedings of the Proceedings of the 2007 Conference on Designing for User eXperiences; 2007; pp. 2–13.

Later, the report gives the impression that there was more than one question in the survey (“Secondly, since all responses were requested to be given in a short format, some of the comments and suggestions given by the students were limited” – l. 300-301). Perhaps including a copy of the survey may contribute to the clarity of the manuscript.

We have included in text explicitly that the surveyed students were answering a unique question (“Secondly, since all responses to the single open-ended question were requested to be given in a short format, some of the comments and suggestions given by the students were limited”). Besides, following your suggestion, we have included a template of the survey presented to the students (Figure 1).

In terms of analysis of data, the authors may want to use emergent themes as an inductive analysis that will allow them to present the results in a condensed format by merging similar responses. As the results are presented in the manuscript, there is too much repetition of responses to the point that it appears that the data was not analyzed using a qualitative approach. In fact, one of the most mentioned topics was that students disliked having their cameras on during video conferencing, but there is no explanation as to why the students felt that way other than “there were privacy concerns.” What did the students mean by that?

We have included in the paper different comments given by the students when explaining their privacy concerns.

There were some explanations presented in the literature review but not from the collected data.

We have reworded some paragraphs, including added connections between the literature review and our findings.

On the other hand, the conclusion, limitations of the study, and further research are clear and inspiring to conduct more research.

Round 2

Reviewer 2 Report

The author addresses in a resonable way the aspects that I mention in previous review.  For me, the design is a mix of qualitative (modality of data collection) and quantitative (reported the results). This is still quite unclear presented in the methodological part of the paper.

Author Response

Thank you very much for your comments.

We have introduced some changes in Section 2 ('Materials and Methods')

  • We have introduced the word ‘qualitative’ to reinforce the approach of the BLA tool

… UX assessment through a qualitative approach, the pocket BLA tool…

  • We have reworded lines 185-192, including additional information:

… Through the BLA tool, different elements of the user experience are named through a description and scored from 0 (lowest possible level of satisfaction) to 10 (maximum level of satisfaction), depending on the experience that each user has lived. According to [97], values close to 0 show that users find the referenced element very uncomfortable or unpleasant, and if it is highly cited, it can be considered a warning that the item needs improvement. However, values close to 10 indicate that users consider an element to be very comfortable or pleasant, and therefore, this item could remain the same. In addition, it can be useful to know the level of heterogeneity of an element, since if a highly cited element has very different marks, we can think that it affects the users in a very different way. Besides, the user is

  • We have included additional information in lines 214-215:

… a specific element, while the average of the score gives an idea of the degree of user satisfaction with the item, and the variance reflects the degree of heterogeneity.